# Cellulose Modification for Improved Compatibility with the Polymer Matrix: Mechanical Characterization of the Composite Material

**DOI:** 10.3390/ma13235519

**Published:** 2020-12-03

**Authors:** Stefan Cichosz, Anna Masek, Adam Rylski

**Affiliations:** 1Institute of Polymer and Dye Technology, Faculty of Chemistry, Lodz University of Technology, Stefanowskiego 12/16, 90-924 Lodz, Poland; stefan.cichosz@p.lodz.pl; 2Institute of Materials Science and Engineering, Lodz University of Technology, 90-924 Lodz, Poland; adam.rylski@p.lodz.pl

**Keywords:** cellulose fibers, ethylene–norbornene copolymer, composites, compatibilization

## Abstract

The following article is the presentation attempt of cellulose hybrid chemical modification approach as a useful tool in improving the mechanical properties of plant fiber-filled polymer materials. The treatment process is a prolonged method of the cellulose maleinization and consists of two steps: 1. solvent exchange (altering fiber structure); 2. maleic anhydride (MA) chemical grafting (surface modification). Thanks to the incorporated treatment method, the created ethylene–norbornene copolymer composite specimen exhibited an improved performance, tensile strength at the level of (38.8 ± 0.8) MPa and (510 ± 20)% elongation at break, which is higher than for neat polymer matrix and could not be achieved in the case of regular MA treatment. Moreover, both the Payne effect and filler efficiency factor indicate a possibility of the fiber reinforcing nature that is not a common result. Additionally, the polymer matrix employed in this research is widely known for its excellent resistance to aqueous and polar organic media, good biocompatibility, and the ability to reproduce fine structures which makes it an interesting material regarding healthcare applications. Therefore, plant fiber-based polymer materials described in this research might be potentially applied in this area, e.g., medical devices, drug delivery, wearables, pharmaceutical blisters, and trays.

## 1. Introduction

Sustainable development is a process which evolves considering different aspects of peoples’ lives throughout the years. Besides being an important part of the social beliefs and governance, it involves multiple transitions or system innovations and changes with technological progress [1]. Thinking about polymer technology is a great challenge is to minimize plastic waste [2,3] and to limit environmental pollution [4,5,6]. The first solid waste management models concentrated on dealing with specific aspects, rather than have being focused around integrated waste management [7]. Only then, the concept of sustainable waste management became central to these models [8].

Rising environmental awareness requires the development of materials which are less harmful to our habitat and take advantage of natural environment around us. In order to compromise the sustainable development rules and polymer technology, it is of a high importance to incorporate materials which are eco-friendly [9,10,11,12,13,14,15] and to limit the noxious chemical compounds employed in polymer processing [16,17,18,19]. Therefore, searching for some natural alternatives seems to be a great option and it is widely being developed [20,21,22,23,24].

In the following research, an application of biodegradable natural fibers in polymer composites is presented. Nevertheless, strongly polarized cellulose is not inherently compatible with commonly used hydrophobic polymer matrices [25]. As a consequence, it requires surface modification such as the employment of various coupling agents which improve the polymer matrix–filler adhesion [26,27,28].

Fortunately, there are known different treatment paths [29]. Physical ones usually lead to the proton abstraction and creation of unstable radicals that may convert chemical groups [30], for example, electric discharge methods (cold plasma treatment [31], corona discharge [32]). These treatments may cause chemical implantation, etching, polymerization or crystallization. Consequently, surface properties of cellulose fibers are altered and the filler–polymer matrix mechanical bonding is improved [25,26].

On the other hand, considering chemical modification, coupling agents facilitate the optimum stress transfer at the matrix–filler interface by cellulose surface hydrophobization (fiber–fiber interaction reduction) or by creating a bridge of chemical bonds between the fiber and the matrix [33,34,35]. Commonly used chemical modification methods are: acetylation [36]; alkali treatment [37]; graft copolymerization [26]; and reactions with isocyanates [38], triazines [39], organosilanes [40] and anhydrides [41].

Another interesting example of chemical modification is polymer grafting, which is a process that has enabled the introduction of well-defined polymer chains of highly controlled structure. This widens the cellulose application range [26,42]. Depending on the grafted polymer type, natural fiber properties such as elasticity, absorbancy, ion exchange capabilities, temperature responsiveness, thermal resistance, abrasion and wear resistance might be affected [43,44,45,46].

This article concentrates on the process of cellulose maleinization and its development. It is the opposite to the commonly used process of polymer matrix maleic anhydride (MA) grafting, and then mixing with cellulose. In this study, an alternative approach, which aims to employ MA not as a direct coupling agent between cellulose and polymer matrix, but as a factor altering the cellulose surface properties, (e.g., hydrophobicity, surface area) is presented. Nonetheless, it should be highlighted that the employment of MA-modified PE (polyethylene) or PP (polypropylene), systems similar to the ethylene–norbornene copolymer, filled with natural fibers provides polymer composites with a good mechanical characterization [47,48,49,50].

The simple modification of the biopolymer with the use of maleic anhydride (MA) was tested before by the authors and the results have been widely described [51]. Moreover, the application of treated cellulose fibers in polymer composites has been examined as well [52].

In spite of the successful esterification of the biopolymer [51], modified cellulose-loaded ethylene–norbornene copolymer composites did exhibit poor mechanical properties (tensile strength of (25 ± 3) MPa and elongation at break at the level of (350 ± 30)%) compared with the performance of the neat polymer matrix (tensile strength of (38 ± 3) MPa and elongation at break (490 ± 30)%) [52]. The modified filler did not ensure appropriate stress transfer on the filler–polymer matrix interface border.

Therefore, this research is a broadening of the previously presented topic and successfully presents the outcome of the improved cellulose treatment process. Here, a combined cellulose modification approach has been introduced—physical treatment was combined with a chemical one. Hybrid chemical modification consists of two steps: solvent exchange (with ethanol either hexane) and chemical grafting (maleic anhydride—MA). MA has been used in order to change the surface characterization of the cellulose fibers and to improve the filler–polymer matrix adhesion. Its positive effect on the composite mechanical properties has been described in the literature [53,54,55]. Thanks to the performed complex modification process, cellulose is pre-treated in order to change its physical features, such as molecular packing, pore sizes and hydrogen bonds organization, which may lead to alterations of the reactivity [56].

Previously, the solvent exchange idea has been proposed by Ishii et al. [57]. It was reported that the presence of solvent particles between cellulose macromolecules loosens the surface fractal of the microfibril aggregate. Consequently, the aggregate geometry is changed into a mass fractal [58]. The general conclusion from the research performed by Ishii et al. was that the solvent exchange improves the cellulose molecular mobility and shortens the characteristic length along the cellulose microfibrils [57]. Therefore, the process of the filler chemical modification described in this research was performed taking advantage of the biopolymer molecular mobility alterations during the solvent exchange. After the complex treatment, cellulose was altered regarding its physical structure (swelling in a solvent) and surface properties (chemical grafting of MA).

Thanks to the presented new modification approach, the polymer composite samples of satisfying mechanical performance have been obtained. After the cellulose modification process, ethylene–norbornene copolymer composites filled with hybrid chemically treated cellulose fibers have been prepared and tested for their mechanical properties. The filler structure development and material stiffness, as well as tensile strength and elongation at break have been assessed with the employment of static and dynamic mechanical analysis methods.

The obtained results are of the great importance considering the possibility of the treated fibers application in polymer matrixes. The newly incorporated modification approach may lead to the broadening of biopolymer application range in polymer technology and the presented results are the perfect confirmation of the cellulose hybrid chemical modification positive impact on polymer composite mechanical properties.

Moreover, the combination of unusual cellulose features with the characteristics of the ethylene–norbornene copolymer may lead to the polymer composite healthcare applications development. The abovementioned polymer matrix is widely known for its resistance to aqueous and polar organic media, as well as good biocompatibility, easy processing and forming. Moreover, the study performed by Wang et al. [59] indicate that MA containing materials have no significant toxic effect regarding the growth of both *Staphylococcus aureus* and *Escherichia coli*. Therefore, plant fiber-based polymer materials described in this research could possibly be applied in medical and healthcare applications, e.g., drug delivery, wearables, different medical devices, pharmaceutical blisters, and trays. Nonetheless, further research regarding the toxicity of maleic anhydride should be carried out in the future.

## 2. Materials and Methods

### 2.1. Preparation of the Specimens

#### 2.1.1. Materials

Arbocel UFC100 Ultrafine Cellulose for Paper and Board Coating from J. Rettenmaier & Soehne (Rosenberg, Germany) was employed in the following research (fiber length: 8–14 µm, density: 1.3 g/cm^3^, pH: 5.0–7.5). Furthermore, the investigated biopolymer was treated with maleic anhydride (MA) provided by Sigma-Aldrich^®^ (Darmstadt, Germany). All experiments were carried out in the presence of various organic solvents: acetone (A), ethanol—99.9% (E) and hexane (H) which were purchased from Chempur^®^ (Piekary Śląskie, Poland). Some crucial properties of the organic solvents are presented in Table 1.

In order to prepare the composite samples, a thermoplastic elastomer, namely ethylene–norbornene copolymer (TOPAS Elastomer E-140 from TOPAS Advanced Polymers, Runheim, Germany) was employed. It is an interesting alternative to traditional flexible materials for use in many areas, such as medical devices, articles for the optical or packaging industry, injection molding.

#### 2.1.2. Cellulose Hybrid Chemical Modification

The major advantage of the presented modification approach is the fact that commonly known chemical treatment of cellulose fibers with MA was broadened with the physical modification (pre-swelling) in media of various polarity and structure (ethanol and hexane) enabling more efficient treatment and improvement in polymer composite performance.

The presented study covers a few problems: effect of the solvent exchange (different polarity of ethanol and hexane) on the properties of cellulose fibers, impact of the dehydration process on the filler modification process, the combined influence of drying, solvent exchange and MA grafting on the properties of both the cellulose and filled polymer composites.

Solvent exchange has been carried out as follows: dispersion mixing (cellulose (g) to solvent (mL) ratio—1:10; magnetic stirrer, 8 h, room temperature) and its conditioning for 16 h in ambient conditions (no stirring). Then, the solvent was distilled in a vacuum rotary evaporator at 40 °C (60 rpm, 100 mbar for ethanol and 250 mbar for hexane).

Modification with maleic anhydride: MA solution with dispersed cellulose (cellulose (g) to acetone (mL) ratio—1:10, cellulose (g) to MA (g) ratio—4:1, 2 h, 40 °C, stirring). Acetone was removed with the vacuum distillation process (40 °C, initial pressure 200 mbar). Further drying occurred in a vacuum oven (100 °C, 440 mbar, 4 h).

Between the steps of solvent exchange and modification with MA, cellulose fibers were stored at 70 °C, and then at 40 °C after finishing whole modification process (before incorporation into the polymer matrix). A summary of the experiments carried out is presented in Table 2.

#### 2.1.3. Polymer Composite Sample Preparation

Modified cellulose fibers were dried for 24 h at 100 °C before incorporation into the polymer matrix. The polymer blends of cellulose (7, 14, 35 wt%) and ethylene–norbornene copolymer were prepared in a laboratory micromixer (Brabender Lab-Station from Plasti-Corder with Julabo cooling system, Duisburg, Germany) at 110 °C (30 min, 50 rpm). For orientation of the fibers, the mixture was put between two-rolling mills (100 × 200 mm rolls, 20–25 °C, friction of 1:1.1, 45 s). Then, it was compressed in a hydraulic press at 160 °C (10 min, 125 bar) between two steel molds and Teflon sheets—the plates were obtained.

### 2.2. Polymer Composite Sample Characterization

Some properties of polymer composites filled with hybrid chemically modified cellulose fibers, as well as the treatment process effect on biopolymer characteristics, are described in separate scientific articles:moisture content effect on properties of cellulose fibers [56] and polymer composites [60],hybrid chemical modification process—biopolymer hydrophobization [61],thermal behavior of hybrid chemically modified cellulose filled polymer blends [62].

#### 2.2.1. Static Mechanical Analysis

Mechanical properties such as tensile strength (TS), elongation at break (Eb) and Young modulus values at elongation equal 100%, 200%, 300% (SE100, SE200, SE300) were determined with the use of Zwick-Roel Z005 (Ulm, Germany) measuring device. Tests were carried out on dumbbell shaped, approximately 1.5 mm thick and 4 mm width specimens, according to PN-ISO 37:1998 standard. The exemplary stress–strain curves are available in the Appendix A.

Moreover, in order to determine the impact of the high-aspect ratio filler orientation within a polymer matrix, samples were cut out in two directions: vertically (|) and horizontally (-). The orientation influence factor (O) has been calculated as follows:(1)$$O=1-\frac{{\mathrm{TS}}_{-}\cdot {\mathrm{Eb}}_{-}}{{\mathrm{TS}}_{|}\cdot {\mathrm{Eb}}_{|}}$$
where:

$TS_{-}$ and $TS_{|}$ are tensile strength of the samples cut out, vertically and horizontally, respectively (MPa).

$Eb_{-}$ and $Eb_{|}$ are the elongation at break of the samples cut out, vertically and horizontally, respectively (%).

#### 2.2.2. Dynamic Mechanical Analysis (DMA)

Both dynamic moduli, storage (E′) and loss (E″), were established with Ares G2-rheometer (parallel plates of 25 mm diameter) from TA Instruments^®^ (New Castle, UK)—soak time of 10 s, angular frequency of 10 rad/s, logarithmic sweep with 0.005–70% strain and 20 points per decade, initial force of 5 N. The experiment was carried out at room temperature. Storage and loss moduli values enabled calculation of the Payne effect (ΔE). The exact equation is given below [63]:(2)$$\Delta E={E^{\prime }}_{\max}-{E^{\prime }}_{\min}$$
where:

${E^{\prime }}_{\max}$, ${E^{\prime }}_{\min}$ are the maximum and minimum value of the storage modulus (MPa).

Additionally, regarding the assessment of cellulose reinforcing effect, the filler efficiency factor (r) has been calculated [64]:(3)$$r=\;\frac{\frac{{E^{\prime }}_{c}}{{E^{\prime }}_{m}}\;-1}{V_{F}}$$
where:

${E^{\prime }}_{c}$ is the storage modulus of filled material (MPa)

${E^{\prime }}_{m}$ is the reference storage modulus (MPa)

$V_{F}$ is the filler volume fraction (-)

## 3. Results and Discussion

### 3.1. Static Mechanical Investigation

Taking into consideration the idea of filling the hydrophobic polymer matrix with hydrophilic natural additive, a significant drop in the material’s performance should be expected. Some research studies explained this phenomenon with the lack of filler–matrix adhesion when unmodified cellulose is incorporated [65]. Nonetheless, the idea of polymer material reinforcing with high aspect ratio polar particles, e.g., natural fibers, is being constantly developed [66,67].

The performed research fulfils these requirements and provides the successfully hydrophobized cellulose fibers for high-performance polymer composite applications, which was achieved with a newly developed hybrid chemical modification.

Data presented in Figure 1a–d indicate that the cellulose treatments carried out exhibited a varied effect on the mechanical properties of the prepared composite samples. It could be observed that many treatments have led to the decrease in composite specimen mechanical performance. According to the literature, it could be caused by insufficient cellulose hydrophobization with the employment of MA, as it was most commonly used in the role of coupling and not a hydrophobizing agent [54,68,69]. Nonetheless, in some particular cases it was possible to regenerate the performance of a pure polymer matrix or even to create a material of an improved tensile strength or elongation at break. This is very promising, and not common according to the information given in the literature [52,70,71,72], and highlights the significance of cellulose structure altering during the solvent exchange step [56].

Considering the available data, although polyethylene, being a similar system to the ethylene–norbornene copolymer, is filled with modified cellulose fibers, its composite tensile strength varies mostly between 25–40 MPa [50,73], depending on the polyethylene source and type. Therefore, it should be underlined that in this research some materials with a performance around 35–38 MPa have been obtained.

Regarding Figure 1a,b, it can be observed that higher tensile strength and elongation at break values are evidenced for the systems filled with treated UFC100 which was not dried prior to the modification process (blue color). Moreover, the most satisfying mechanical results are visible for the treatments that involved ethanol employment in a solvent exchange stage. It may have been caused by some specific solvent–cellulose interactions [57,74] and the hornification process [75,76] occurring upon the modification.

In spite of being diagnosed in terms of the physical changes occurrence, the hornification phenomenon, in its origin, has frequently been associated with the formation of irreversible or partially irreversible hydrogen bonds in plant fibers upon drying or water removal. However, it is also considered that it may be connected with lactone bridges formation [77].

Considering the MA treatment of undried fibers, regular maleinization of cellulose caused a significant decrease in tensile strength. Nevertheless, when either ethanol or hexane are incorporated into the modification process, then the composite sample performance increases. This leads to the conclusion that the hybrid chemical modification is more efficient.

On the other hand, when giving a look at the systems filled with UFC100 dried prior to the chemical treatment, one might perceive an opposite situation. The material performance drops while modification process is broadened with the solvent exchange stage.

It might be explained by the fact that the discussed hornification process may not only lead to the strengthening of natural fibers, but also to the blocking of hydroxyl groups and prohibiting the chemical grafting of MA. Consequently, only partially modified fibers are obtained [56,78]. Therefore, it is not desired to dry the cellulose fibers before the modification process. This problem has been widely described in our previous work [61].

What is surprising, according to Figure 1c,d, when only solvents are applied in the modification process of natural fibers, then an opposite effect of the cellulose drying process may be detected. For not dried cellulose, only a slight impact on composite sample tensile strength and elongation at break is observed. However, while considering UFC100 dried before the solvent exchange, the largest improvement in mechanical properties is evidenced. This could be explained by the further hornification of pre-dried cellulose in different solvents, as more wetting–drying cycles in a specific environment are employed. Therefore, the fiber becomes stiffer [79].

Furthermore, the discussed improvement in tensile properties has been mostly investigated regarding the tensile strength and elongation at break results evidenced only for one cellulose alignment direction (|). However, as natural fibers are high aspect ratio particles [80], it is strongly recommended that the mechanical characteristics should be also determined in an opposite, perpendicular direction (-). Therefore, orientation influence factor, which helps to assess differences in tensile behavior of polymer composite samples, has been calculated (Figure 1e).

The lower the orientation influence factor value, the less significant the direction of cutting out of the samples, from the tensile properties point of view. This may indicate some information about the filler–polymer matrix adhesion improvement, as the cellulose fibers alignment impact decreases.

Taking into consideration the data presented in Figure 1a–e, the composite specimen exhibiting the best relative mechanical performance in both directions may be chosen. It could be concluded that the most satisfying results have been obtained for the cellulosic filler which was not dried prior to the modification process with MA preceded by the solvent exchange to ethanol (ND/MA/1E).

Moreover, an interesting trend is presented in Figure 1f. For all investigated composite samples filled with cellulose fibers, a higher tensile strength was observed in the case of specimens exhibiting an elevated elongation at break. This means that no significant material stiffening was observed.

However, this detected phenomenon is not common for polymeric materials [50,81,82] and could be caused by unusual properties of the ethylene–norbornene copolymer, sometimes referred to as a thermoplastic elastomer (due to the presence of elastic and rigid blocks) [44,63].

In summary, considering the hybrid chemical modification, the most favorable changes were evidenced for cellulose not dried before the treatment and for the employment of ethanol in the solvent exchange stage. On the other hand, when only a solvent treatment was applied, fibers which were subjected to more drying–wetting cycles, resulted in the elevated polymer composite improved performance. Most of the observed changes could originate from the hornification process of natural fibers which is altered by the solvents of different polarity. Furthermore, the performed treatments highly affected the cellulose orientation influence factor, which may be caused by some specific filler–polymer matrix interactions and the blocking of hydroxyl groups in cellulose.

### 3.2. Dynamic Mechanical Analysis

Dynamic mechanical analysis may provide some valuable information considering rheology properties, e.g., viscosity, storage (E′) and loss (E″) moduli, and Payne effect [83,84,85,86]. Here, it was carried out in order to assess the bio-filler reinforcing nature and support the tensile test results. Therefore, attention was mostly paid to the dynamic moduli values and their changes with the oscillation strain.

Oscillation may destroy the filler structure developed within the polymer matrix and on the basis of dynamic moduli changes occurring during its destruction, the filler structure development degree may be calculated.

In Figure 2a, the graph illustrates the variations of the analyzed composites’ storage and loss moduli upon oscillation changes. A typical Payne effect might be detected. While E″ achieves its maximum value, E′ falls down [87] and the damping factor (tan δ) value increases. According to the literature, the Payne effect might be assigned to the destruction of the filler structure developed within analyzed polymer matrix [88].

According to the literature, varied dynamic mechanical properties of polymer composite sample may be explained by the wetting degree of the fiber with the matrix, uneven aligning of cellulose fibers, and filler–polymer matrix adhesion problems. Consequently, this may lead to the formation of numerous voids at the fiber-matrix interface. Therefore, the stress transfer to the fibers, which are the load bearing entities, becomes inefficient [83,89].

Firstly, analyzing data presented in Figure 2b,c, it could be concluded that almost all of the performed modifications led to a significant rise in storage modulus and a slight decrease in loss modulus values, compared with the neat cellulose fibers’ incorporation into the ethylene–norbornene copolymer.

What should be underlined, in some particular cases, when cellulose has been modified with ethanol and cellulose fibers were not dried prior to the modification process, the improvement in dynamic moduli has exaggerated values assigned to the neat polymer matrix.

This provides some information about the filler–polymer matrix interface properties. Higher storage modulus may mean more intense interactions between the two abovementioned components, providing improved mechanical properties [90]. On the other hand, while comparing the samples modified with only solvents and with maleic anhydride, it could be observed that a higher storage modulus is detected in the case of the first mentioned group, i.e., ND/1/E compared to ND/MA/0. This might have been caused by the cellulose structural changes occurring upon the solvent exchange process and fiber stiffening.

Figure 2d reveals the Payne effect (ΔE) and filler efficiency factor (r) values calculated regarding all analyzed composite samples. It can be seen that the modifications performed with the employment of solvent exchange to ethanol give a higher Payne effect and filler efficiency factor values. This is in a good correspondence with the obtained tensile test results.

Moreover, comparing the effect of the cellulose drying during the modification process, it may be observed that for UFC100/ND incorporation to polymer matrix, the Payne effect varies between 0.43 MPa and 1.60 MPa, while in the case of UFC100/D, it is in the range from 0.84–1.26 MPa. Therefore, fibers dried prior to the further chemical modification are supposed to allow a more regular and less developed filler structure in the polymer matrix [91], compared to UFC100/ND addition (for UFC100/D: the results range is narrower and the maximum possible value is lower).

In order to understand ongoing changes, a number of theories explaining the Payne effect have been proposed, e.g., particles clustering into the network or physically jammed domains via interfacial interactions between filler and matrix [91]; molecular disentanglement accelerated by particles subjected to the straining forces; and the rubbery phase undergoes a microscopic strain higher than the macroscopic strains applied on the compounds [92].

Additionally, a well-developed filler structure is not always in good correlation with composite performance improvement—filler aggregation may be evidenced and misunderstood as a good filler distribution within the polymer matrix [93].

Fortunately, another factor indicating filler behavior within the polymer matrix is established. The filler efficiency factor (r) compares the storage modulus of the unfilled and filled system regarding the filler volume (which was the same in all cases).

Therefore, regarding data given in Figure 2d, it could be observed that the filler efficiency factor is positive only for three composite samples all of which were filled with cellulose not dried before the modification process. One of them is the specimen highlighted before—not dried cellulose fibers modified with MA after a solvent exchange to ethanol (ND/MA/1/E). According to gathered results, this is the sample exhibiting the best mechanical properties.

In conclusion, the reinforcement of the polymer matrix is a more complex phenomenon. Besides being dependent on the good filler distribution and structure development within the polymer matrix, also other factors are of a high importance, e.g., filler–polymer matrix adhesion, interface properties, and possible entanglements [94].

Furthermore, it may be seen that the most satisfying results are detected considering the samples modified only with maleic anhydride or in the case of ethanol employment during the solvent exchange step in the modification process (for undried cellulose). The rest of the analyzed composite samples did not exhibit the positive value of r factor. Therefore, there is no storage modulus improvement evidenced while comparing with the neat polymer matrix.

What should be underlined, the observed reinforcing effect according to literature, is the overall impact of two different mechanisms occurring simultaneously—filler strengthening the polymer matrix, as well as interfacial bonding between the cellulose and ethylene–norbornene copolymer [90].

### 3.3. Comparison of the Obtained Results with Literature

#### 3.3.1. Tensile Properties

Considering the data gathered above, one type of cellulose fibers has been chosen for further analysis: undried UFC100 treated with MA after the solvent exchange to ethanol (ND/MA/1/E).

According to the obtained results, ND/MA/1/E was the cellulose sample exhibiting the lowest moisture content, which was examined before [61], and its incorporation into ethylene–norbornene copolymer resulted in the improved mechanical properties of polymer composite specimen (relatively low orientation influence factor, rise of tensile strength and elongation at break, improved dynamic moduli, high Payne effect value and positive filler efficiency factor). Therefore, results presented from that moment on concern the ND/MA/1/E type of cellulose (Figure 3). In Figure 3a, the effect of the treated filler amount in the polymer matrix is revealed. By altering the polymer composite recipe and lowering the cellulose loading from 14 wt% to 7 wt%, the improvement in the tensile strength and elongation at break may be obtained. What should be underlined is, in this way not only was the performance of neat polymer matrix restored, but also a material of a higher tensile strength and elongation at break has been obtained.

On the other hand, 35 wt% of cellulose causes the degradation of material mechanical properties. Here, the filler might create the clusters which do not transfer the tension within the polymer composite as it was not efficiently wetted by the polymer matrix [95].

Moving forward, cellulose content up to 14 wt% had no significant influence on the tension values at elongations of 100%, 200% and 300% (Figure 3b). Nonetheless, while 40 phr of cellulose were incorporated into ethylene–norbornene copolymer, the Young moduli decreased and the specimen broke before achieving 300% elongation.

Furthermore, we would like to show how the material created in this research is different from various cellulose-based polymer composites presented in the literature. Therefore, some additional data was gathered in Table 3. It helps to compare the results obtained in this research study with the performance of polymer composites prepared by different scientists.

It is clearly visible that, considering tabularized data, the tensile strength of ethylene–norbornene copolymer-based composites is quite high and is only slightly lower than for the nanocellulose-filled HDPE sample. However, after closely examining the elongation at break values, it can be observed that the results obtained for specimens obtained in this research are the highest (regarding available data).

Moreover, analyzing the information presented in Table 3, in most cases elongation at break decreases with the modified bio-filler incorporation to the polymer matrix. Nevertheless, an opposite effect might be observed regarding the system proposed in this research (Figure 1f).

This phenomenon could be explained regarding the extraordinary properties of the ethylene–norbornene copolymer which are the effect of elastic ethylene segments and rigid norbornene blocks combination [96]. However, it could be also due to the improved filler–polymer matrix interactions considering the cellulose hydrophilization process carried out [28,78,97,98].

Therefore, it could be claimed that this research study provides some valuable data considering the mechanical performance of the cellulose-based material which is reinforced with the added filler (in an appropriate amount). The obtained results differ from the information presented in the literature.

#### 3.3.2. Dynamic Mechanical Analysis

Dynamic mechanical analysis is referred to as a perfect method to reveal the facts about the heterogenous polymeric systems and polymeric composite materials [101]. It may provide a wide range of information concerning the filler dispersion, material stiffness, and damping behavior of the polymer composites [102].

Interesting results have been presented by Shumigin et al. [103]. The research compares the impact of untreated cellulose fibers on poly(lactic acid) (PLA) and low density polyethylene (LDPE). The authors evidenced the higher melt viscosity when the polymer matrix was loaded with cellulose. Moreover, it was reported that the higher the volume fraction and droplet size of cellulose phase, the higher the E′, especially at low angular frequencies. This means dynamic properties may be significantly affected by the amount of the bio-filler.

Similarly, Saba et al. [101] reported that the storage modulus increases with a weight fraction of cellulose fibers due to the increase in stiffness conveyed by the strong interactions developed between the filler and polymer matrix. On the other hand, the increase in cellulose fibers causes a drop in the damping nature of the composite because the amplified stiffness is imparted by the cellulose aggregates.

The study, being the subject of this article, reveals similar trends which are in accordance with the works gathered in Table 4. While the composite sample tensile strength reinforcement is observed, an increase in the storage modulus may be also detected. Nevertheless, this effect is not evidenced concerning all of the polymer composite samples filled with the modified cellulose fibers. Therefore, the method of cellulose modification varies the final material properties to different extents. Lower values of storage and loss moduli might evidence the poor filler–polymer matrix interactions [104,105] and, therefore, the low modification yield.

## 4. Conclusions

This article reveals the usefulness of the performed cellulose hybrid chemical modification in improving the mechanical properties of plant fiber-filled polymer composites. What should be emphasized is that the method is simple in its principle and gives better results, considering the improvements in material performance in comparison with one-step cellulose maleinization process.

The mechanical properties of the eco-friendly polymer composites filled with hybrid chemically modified cellulose fibers have been analyzed with the employment of tensile and dynamic tests. Taking into consideration all of gathered results, it may be said that regarding mechanical properties:the highest performance was noticed in the case of the sample filled with ND/MA/1/E cellulose; here, undried cellulose was modified with MA after a solvent exchange with ethanol and the biopolymer was not dried before the carried out treatment;reinforcement of ethylene–norbornene copolymer with the employment of modified cellulose fibers has been successfully achieved—up to TS = (38.8 ± 0.8) MPa and Eb = (510 ± 20)%.

Moreover, materials presented in this research are incredibly promising, considering an opportunity of bio-based polymeric composites’ employment in common healthcare applications, as the ethylene–norbornene copolymer is widely used in this area. The polymer matrix is characterized by high purity, excellent barrier properties, and it can be sterilized by all known methods. Furthermore, its easy production enables forming opportunities which were not available for glass products (mostly used in the past). Nevertheless, this material needs to be well optimized in the future.

## Figures and Tables

**Figure 1 materials-13-05519-f001:**
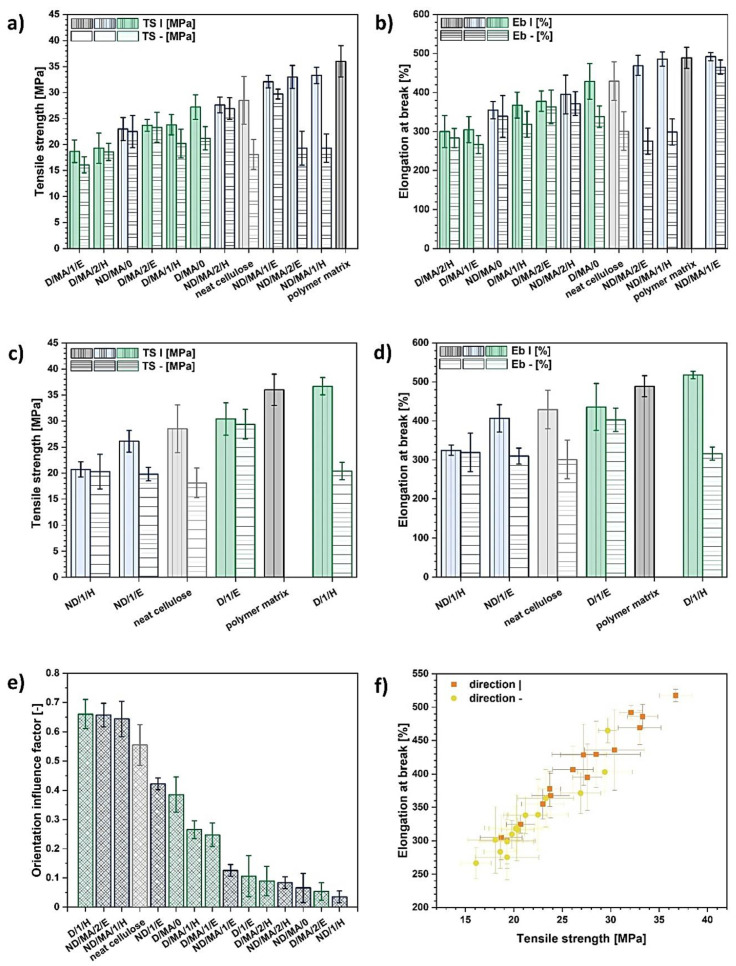
Tensile properties comparison: tensile strength and elongation at break of hybrid chemically modified cellulose-filled (**a**,**b**) and solvent-treated cellulose-filled (**c**,**d**) polymer composites, orientation influence factor (**e**), elongation at break as a function of tensile strength (**f**) for all investigated composite samples; TS, tensile strength (MPa); Eb, elongation at break (%); |,- direction of cutting out of the specimens. Cellulose loading: 14 wt%.

**Figure 2 materials-13-05519-f002:**
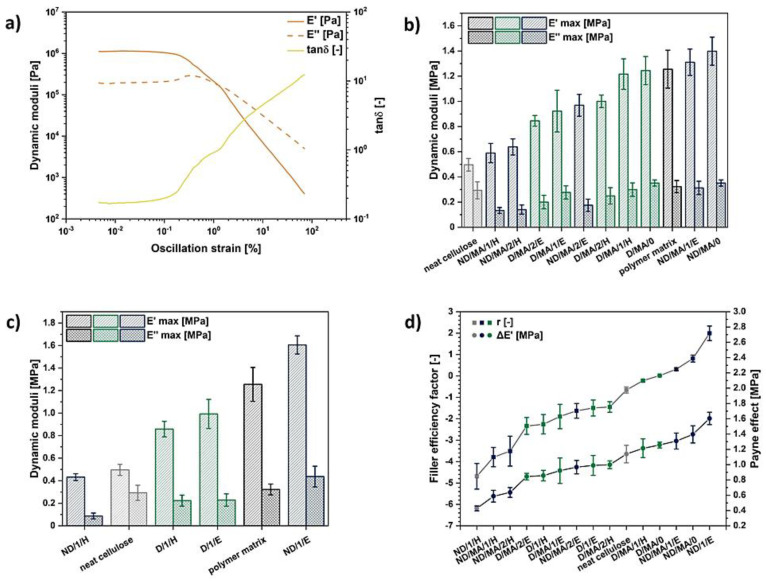
Dynamic mechanical analysis results: an example of the curve (ND/MA/0) with visible Payne effect (**a**); the maximum values of storage and loss moduli for composite samples filler with hybrid chemically modified cellulose (**b**) and solvent treated natural fibers (**c**); filler efficiency factor (r) and Payne effect values (ΔE) for different treatments (**d**); E′, storage modulus (MPa); E″, loss modulus (MPa); tan δ, damping factor (-). Cellulose loading: 14 wt%.

**Figure 3 materials-13-05519-f003:**
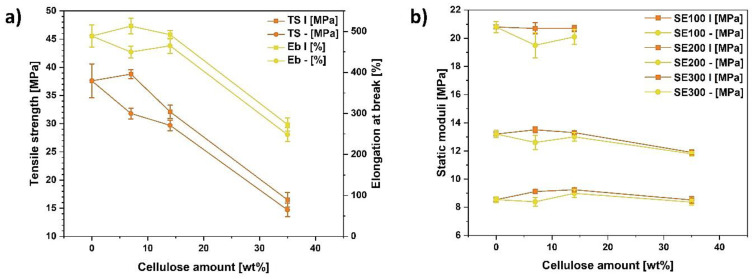
Tensile properties comparison presented as a function of modified cellulose fibers (ND/MA/1/E) amount: (**a**) tensile strength (TS) and elongation at break (Eb); (**b**) moduli (SE) at elongations of 100%, 200% and 300%.

**Table 1 materials-13-05519-t001:** Physical properties of solvents employed in experiments.

Property	Acetone	Ethanol (99.9%)	Hexane
boiling point (°C)	55–57	78	68
solubility in water (g/cm^3^)	yes	yes	0.1
solubility in organic solvents	yes	yes	yes

**Table 2 materials-13-05519-t002:** Summary of all performed cellulose modifications; H—hexane, E—ethanol, MA—maleic anhydride.

Sample	Dried before Modification (D)	Not Dried before Modification (ND)	Solvent Exchange				MA Treated
			Before MA Treatment		After MA Treatment		
			H	E	H	E	
ND/MA/0	-	✔	-	-	-	-	✔
ND/MA/1/H	-	✔	✔	-	-	-	✔
ND/MA/1/E	-	✔	-	✔	-	-	✔
ND/MA/2/H	-	✔	-	-	✔	-	✔
ND/MA/2/E	-	✔	-	-	-	✔	✔
ND/1/H	-	✔	✔	-	-	-	-
ND/1/E	-	✔	-	✔	-	-	-
D/MA/0	✔	-	-	-	-	-	✔
D/MA/1/H	✔	-	✔	-	-	-	✔
D/MA/1/E	✔	-	-	✔	-	-	✔
D/MA/2/H	✔	-	-	-	✔	-	✔
D/MA/2/E	✔	-	-	-	-	✔	✔
D/1/H	✔	-	✔	-	-	-	-
D/1/E	✔	-	-	✔	-	-	-

**Table 3 materials-13-05519-t003:** Mechanical properties of cellulose filled polymer composites comparison. Abbreviations: TS, tensile strength; Eb, elongation at break; PE, polyethylene; LDPE, low density polyethylene; HDPE, high density polyethylene; MA, maleic anhydride; EO, ethylene oxide; EPI, epichlorohydrin; NMP, *N*-Methyl-2-pyrrolidon; CNC, cellulose nanocrystals.

Filler	Modifying Agent/Treatment	Polymer Matrix	Neat Polymer Matrix		Polymer Matrix Filled with Unmodified Bio-Filler		Polymer Matrix Filled with Modified Bio-Filler		Ref.
			TS (MPa)	Eb (%)	TS (MPa)	Eb (%)	TS (MPa)	Eb (%)	
wood flour	ternary-graft copolymers (PE solid phase grafting: graft degree of 10.5%)	recycled PE	20 ± 2	-	-	-	36 ± 2	-	[81]
cotton fabric waste microfibers	chlorination (2 M HCl, 80 °C, 4 h) and further NaOH treatment (60 °C, 4 h)	recycled PE + PE-g-MA	5 ± 1	-	25 ± 4	-	25 ± 3	-	[68]
nanocellulose	alkenyl succinic anhydride (1 h, 70–80 °C in the presence of: K_2_CO_3_, NMP)	HDPE	~21	>10	~5	>10	~43	4	[50]
powdered cellulose	-	LDPE	~7	>20	12	>20	-	-	[99]
clean oil palm empty fruit bunch fibers (holocellulose)	Chlorination (NaCl and CH_3_COOH, 70 °C, 6 h), mercerization (17.5% NaOH, 20 °C, 2 h)	LDPE	~11	~600	-	-	~13	<50	[100]
nanocellulose	PVA (solution casting from DMSO; PVA solution added to EO/EPI/CNC mixture at 70 °C, further sonification)	EO-co-EPI	-	-	0.50 ± 0.07	39.9 ± 0.9	1.98 ± 0.08	17.1 ± 1.4	[82]
THIS RESEARCH STUDY									
cellulose (8–14 μm)	maleic anhydride	TOPAS	36 ± 3	490 ± 20	29 ± 5	430 ± 50	38.8 ± 0.8	510 ± 20	-

**Table 4 materials-13-05519-t004:** Dynamic mechanical analysis of polyolefin composites filled with natural fibers.

Polymer Matrix	Filler	Ref.
high density polyethylene (HDPE)	hemp fibers	[106]
	kenaf bast fiber	[102]
	cotton nanocellulose fiber	[49]
linear low density polyethylene (LLDPE)	cellulose nanocrystal (CNC)	[107]
polypropylene (PP)	orange wood fibers	[108]
	hemp fibers	[104]
	cellulose nanocrystals	[109]

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
