# Peer review of "Cellulose Modification for Improved Compatibility with the Polymer Matrix: Mechanical Characterization of the Composite Material"

_materials, 2020, doi:10.3390/ma13235519_

Round 1

Reviewer 1 Report

The paper is focused on the mechanical properties of plant fiber-filled polymer composites. The topic falls within the scope of the journal. The MS can be published after the following revisions:

  • Conclusions paragraph should be shortened.
  • Some examples of stress vs strain curves should be presented.
  • I suggest to estimate the stored energy up to breaking by the analysis of stress vs stain curves.
  • Introduction might me improved by quoting recent articles [New Journal of Chemistry 2019, 43, 10887-10893] and reviews [Carbohydrate Polymers 245 (2020) 116502] focused on the development of bio-composite materials for healcare purposes.

Author Response

Institute of Polymer and Dye Technology

Technical University of Lodz

90-924 Lodz, ul Stefanowskiego 12/16, Poland

Tel.: +48 42 631 32 23, Fax: +48 42 636 25 43

November 12, 2020

Materials

Dear Professor,

We are resubmitting our revised paper entitled Mechanical Characterization of Sustainable High-Performance Polymer Bio-Composites for Potential Healthcare Applications (then: Cellulose Modification for Improved Compatibility with the Polymer Matrix: Mechanical Characterization of the Composite Material) by Stefan Cichosz, Anna Masek with a request to reconsider it for publication in Materials.

We have carefully considered the Editor and Reviewers' comments. The manuscript was revised exactly according to these comments. The list of responses to the reviewer’s comments and corrections made in the manuscript is attached.

The manuscript has not been previously published, is not currently submitted for review to any other journal, and will not be submitted elsewhere before a decision is made by this journal.

For correspondence please use the following information:

corresponding author: Anna Masek

Institute of Polymer and Dye Technology

Technical University of Lodz

90-924 Lodz, ul Stefanowskiego 12/16, Poland

Tel.: +48 42 631 32 93

Fax: +48 42 636 25 43

e-mail: anna.masek@p.lodz.pl

Yours sincerely,

Ph. D., D.Sc. Anna Masek

All changes are marked with a green colour through whole manuscript.

Reviewer #1

The need for eco-friendly materials is explained as well as the cellulose polarisation as an obstacle for its incorporation of polymer matrices. Available surface modification methods of cellulose are described, highlighting the grafting as the most promising one.

Need for current research is justified by the need of improved properties of modified-cellulose loaded ethylene-norbornene copolymer composites, which is also the work of the same authors. The only innovation proposed in the introduction is the solvent exchange before chemical process. Without looking on the “big picture” of all related  papers published by the same authors recently, it seems low novelty potential for this individual manuscript.

In overall, several theories have been proposed and applied to purposefully explain debatable results of effect of modified cellulose on the properties of polymer matrix. Seems quite challenging, however proposed filler efficiency factor and Payne effect explain the results more understandable.

Significant work done on explanation and interpretation of numbers obtained, which resulted in choosing the most promising modification method for further research.

The justification for the need of the particular research  should be improved in the text (introduction), considering this paperwork as individual and not only emphasize it as the continuation of previous work done. The fact that the reader may do not have previous knowledge or information about the specific topic must be considered as well, giving more overall information in the text, not only by referring to previous papers.

The comments are listed below:

  1. More justification of using specifically maleic acid for grafting the cellulose surface is needed in the introduction. What are the advantages of using cellulose maleinization? Which other cellulose like polymers have been grafted and investigated/published using this method?

Answer: We are grateful for this comment. Introduction was improved and necessary information was given: MA has been used in order to change the surface characterization of the cellulose fibers and to improve the filler-polymer matrix adhesion. Its positive effect on the composite mechanical properties was described in literature [53–55]. Thanks to the performed complex modification process, cellulose is firstly pre-treated in order to change its physical features, such as molecular packing, pore sizes and hydrogen bonds organization, which may lead to the reactivity altering [56].

  1. Is there any proof about non-toxicity of cellulose, when it is modified via maleinization? It should be assessed before proposing for application in medical or healthcare applications.

Answer: We are thankful for drawing our attention to this problem. The manuscript has been improved according to the reference: Moreover, the study performed by Wang et al. [59] indicate that MA containing materials have no significant toxic effect regarding the growth of both S. aureus and E. coli. Therefore, plant fiber-based polymer materials described in this research might be possibly applied in medical and healthcare applications, e.g., drug delivery, wearables, different medical devices, pharmaceutical blisters and trays. Nonetheless, further research regarding the toxicity of maleic anhydride should be carried out in the future.

  1. Graphical abstract of manuscript would improve the manuscript.

Answer: We believe that this is a mistake as we have already provided the graphical abstract.

  1. It is great to have summary table of performed modifications, however there is no detailed description of cellulose modification method. Reaction conditions, amounts of used chemicals etc should be added. In my opinion it is not enough to give reference to authors’ previous works. Method must be briefly described also in the current manuscript.

Answer: We are thankful for this comment. The description of the modification method has been incorporated into the Table 3.

  1. The question “What was the size of used cellulose fibres?” occurred during reading the first part of manuscript and it was answered later in Table 3. Considering small sizes of cellulose particles, why do you still call them fibres instead of cellulose microparticles or microcrystalline cellulose (MCC)? Measurements of crystallinity would reveal, if it is MCC. Were there some visualisation methods used (SEM, AFM) in order to reveal shape of cellulose particles?

Answer: Yes, we have done AFM visualisation. Obtained results were published in 2019:  S. Cichosz, A. Masek, K. Wolski, Innovative cellulose fibres reinforced ethylene-norbornene copolymer composites of an increased degradation potential, Polymer Degradation and Stability, 2019, 159, 174-183.

We have not examined the cellulose samples regarding their crystallinity index yet. Nonetheless, we are planning to do so. Therefore, we call it ‘fibres’ instead of MCC. Moreover, the available details regarding the cellulose employed in this research were given in the Materials section: Arbocel UFC100 Ultrafine Cellulose for Paper and Board Coating from J. Rettenmaier & Soehne (Rosenberg, Germany) was employed in the following research (fiber length: 8-14 µm, density: 1.3 g/cm3, pH: 5.0–7.5).

  1. Line 134 Methodology of cellulose fibers drying process has been given, however there is no details about incorporation of non-dried fibres in polymer matrix. Was it the same procedure described in Lines 135 – 141. Did non-dried fibers act the same way as dried in the composite forming process?

Answer: Every cellulose sample has been dried before incorporation into the polymer matrix. The only difference is the drying/non-drying at the modification stage.  Therefore, we have underlined the presence of cellulose fibres drying and the modification process, as well as composite sample preparation description have been improved: The major advantage of the presented modification approach is the fact that commonly known chemical treatment of cellulose fibers with MA was broadened with the physical modification (pre-swelling) in media of various polarity and structure (ethanol and hexane) enabling more efficient treatment and improvement in polymer composite performance. The presented study covers a few problems: effect of the solvent exchange (different polarity) on the properties of cellulose fibers, impact of the dehydration process on the filler modification process, the combined influence of drying, solvent exchange and MA grafting on the properties of both the cellulose and filled polymer composites. Solvent exchange has been carried out as follows: dispersion mixing (cellulose [g] to solvent [mL] ratio – 1:10; magnetic stirrer, 8 h, room temperature) and its conditioning for 16 h in ambient conditions (no stirring). Then, the solvent was distilled in a vacuum rotary evaporator at 40°C (60 rpm, 100 mbar for ethanol and 250 mbar for hexane). Modification with maleic anhydride: MA solution with dispersed cellulose (cellulose [g] to acetone [mL] ratio – 1:10, cellulose [g] to MA [g] ratio – 4:1, 2 h, 40°C, stirring).  Acetone was removed with the vacuum distillation process (40°C, initial pressure 200 mbar). Further drying: vacuum oven (100°C, 440 mbar, 4 h). Between the steps of solvent exchange and modification with MA, cellulose fibers have been stored at 70°C and then after finishing whole modification process – at 40°C (before incorporation into the polymer matrix). Summary of the carried out experiments are presented in Table 2.

Modified cellulose fibers were dried for 24 h at 100°C before incorporation into the polymer matrix. The polymer blends of cellulose (7, 14, 35 wt%) and ethylene-norbornene copolymer were prepared in a laboratory micromixer (Brabender Lab-Station from Plasti-Corder with Julabo cooling system, Duisburg, Germany) at 110°C (30 min, 50 rpm). For orientation of the fibers, the mixture has been put between two-rolling mills (100x200 mm rolls, 20–25°C, friction of 1:1.1, 45 seconds). Then, the it was compressed in a hydraulic press at 160°C (10 min, 125 bar) between two steel molds and Teflon sheets – the plates were obtained.

  1. It is confusing – is the drying step described in Line 134 the same included in Table 1? So when exactly the celullose was dried? Before/after the modification AND before incorporation into polymer matrix? Does it mean that some of samples were dried several times during the whole process? Detailed description of modification process, even graphical image with scheme, would make the methodology more clear and there would be no place for confusion of the reader.

It should be taken into account that if cellulose material was delivered as dry powder, it has already had some wetting- drying cycles. In order to completely exclude hornification effect, never-dried fibres should be used.

Answer: We agree with this comment. The Materials & Methods Section has been improved significantly as shown before.

  1. Figure 1. The images give information about neat cellulose as better filler than modified one in some cases? Could you please explain this fact?

Answer: We are thankful for this comment. We have explained it a little in the manuscript. Nevertheless, the description was developed and improved, e.g.,: Data presented in Fig. 1a-d indicate that carried out cellulose treatments exhibit a varied effect on the mechanical properties of prepared composite samples. It could be observed that many treatments have led to the decrease in composite specimen mechanical performance. According to literature, it could be caused by insufficient cellulose hydrophobization with the employment of MA as it most commonly used in the role of coupling and not hydrophobizing agent [54,68,69]. Nonetheless, in some particular cases it was possible to regenerate the performance of pure polymer matrix or even to create a material of an improved tensile strength or elongation at break. This is very promising and not common according to the information given in literature [52,70–72] and highlights the significance of cellulose structure altering during the solvent exchange step [56].

Reviewer 2 Report

Title

  1. The title does not clearly express the work done in the article. The reviewer suggests that it should be modified. It is suggested to focus the title on the modification of the natural fibers for their compatibility with a plastic matrix.

Abstract

  1. Line 10. The authors say," The following article is the first presentation attempt of cellulose hybrid chemical modification approach as a useful tool in improving the mechanical properties". The authors should clarify the term "hybrid chemical modification". They are performing a combined treatment: physical treatment + chemical treatment? This is a very brave statement; the reviewer has been able to verify with a quick search on the web the science more than 50 articles on this subject.
  2. Line 22. The authors say " Therefore, plant fiber-based polymer bio-materials described in this research". According to the authors, cellulose fibers are the reinforcement and the polymer used is ethylene. Ethylene is not a bio-material. The authors should talk about composite materials reinforced with natural fibers. Only the percentage of reinforcement will be bio.

Keywords

  1. The authors must eliminate bio-composites, because only a fraction of the compound is bio.
  2. The authors must eliminate healthcare, although the material can be applied in this sector, the article does not evaluate its properties in these applications.
  3. Line 42. The authors say " In the following research, an application of fully biodegradable natural fibers is presented". First, using the term fully biodegradable natural fibers is redundant, by definition, natural fibers are fully biodegradable. Secondly, this sentence does not provide any information since the authors would rather be referring to paper production.
  4. Next, the authors make a state of the art on methods of compatibilization natural fibers - plastic matrix. However, they omit the most used method and with better results, the use of coupling agents (plastic polymer with maleic anhydride). The authors should extend the introduction in this sense.
  5. The authors justify the novelty and relevance of the article on the basis that this method allows the use of natural fibers in the production of composite materials. However, there are numerous works where coupling agents type MAPP or MAPE are used, which are much easier and have better performance in terms of improvement of mechanical properties.

Materials and Methods

  1. Although the authors have previously published the method of modifying the surface of the fibers, they must explain step by step the modification of these including all the conditions of the process.
  2. The authors do not specify the meaning of H and E in table 2.
  3. Line 135. What does phr mean? Are they percentages by weight? In volume? What do 8, 16 and 40 answer? Why have these percentages been chosen?
  4. Lines 137-138. The authors say "". In the mill they orient fibers once the composite is produced?
  5. The authors must at least explain on what are based the methods used (gravimetry, etc...) and under which norms the tests have been conducted.

Results and discussion

  1. The authors should know the importance of the type of fibers used. A chemical and morphological characterization of Arbocel fibers is essential to understand their behavior in the modification and inside the composite material. They should include: Lignin content, hemicelluloses, cellulose, average length, average diameter, length distribution, polarity, etc.
  2. The authors base the article on the modification of the fibers and do not present any results: polarity variation, cationic demand, IR, etc.
  3. The graphics of the figures are extremely difficult to read.
  4. The incorporation of cellulose fibers reduces the properties of the matrix, which indicates an ineffective modification of the fibers.
  5. In summary, the article needs to be completely revised to be published.

Author Response

Institute of Polymer and Dye Technology

Technical University of Lodz

90-924 Lodz, ul Stefanowskiego 12/16, Poland

Tel.: +48 42 631 32 23, Fax: +48 42 636 25 43

November 12, 2020

Materials

Dear Professor,

We are resubmitting our revised paper entitled Mechanical Characterization of Sustainable High-Performance Polymer Bio-Composites for Potential Healthcare Applications (then: Cellulose Modification for Improved Compatibility with the Polymer Matrix: Mechanical Characterization of the Composite Material) by Stefan Cichosz, Anna Masek with a request to reconsider it for publication in Materials.

We have carefully considered the Editor and Reviewers' comments. The manuscript was revised exactly according to these comments. The list of responses to the reviewer’s comments and corrections made in the manuscript is attached.

The manuscript has not been previously published, is not currently submitted for review to any other journal, and will not be submitted elsewhere before a decision is made by this journal.

For correspondence please use the following information:

corresponding author: Anna Masek

Institute of Polymer and Dye Technology

Technical University of Lodz

90-924 Lodz, ul Stefanowskiego 12/16, Poland

Tel.: +48 42 631 32 93

Fax: +48 42 636 25 43

e-mail: anna.masek@p.lodz.pl

Yours sincerely,

Ph. D., D.Sc. Anna Masek

Reviewer #2

The comments are listed below:

  1. The title does not clearly express the work done in the article. The reviewer suggests that it should be modified. It is suggested to focus the title on the modification of the natural fibers for their compatibility with a plastic matrix.

Answer: We have corrected the title as requested: Cellulose Modification for Improved Compatibility with the Polymer Matrix: Mechanical Characterization of the Composite Material.

  1. Line 10. The authors say," The following article is the first presentation attempt of cellulose hybrid chemical modification approach as a useful tool in improving the mechanical properties". The authors should clarify the term "hybrid chemical modification". They are performing a combined treatment: physical treatment + chemical treatment? This is a very brave statement; the reviewer has been able to verify with a quick search on the web the science more than 50 articles on this subject.

Answer: We are terribly sorry for this misunderstanding. We would like to ensure that we have never wanted to mislead the reader. Therefore, the sentence was changed as follows:  The following article is the presentation attempt of cellulose hybrid chemical modification approach as a useful tool in improving the mechanical properties of plant fiber-filled polymer materials.

  1. Line 22. The authors say "Therefore, plant fiber-based polymer bio-materials described in this research". According to the authors, cellulose fibers are the reinforcement and the polymer used is ethylene. Ethylene is not a bio-material. The authors should talk about composite materials reinforced with natural fibers. Only the percentage of reinforcement will be bio.

Answer: We are grateful for drawing our attention to this problem. The term “bio-materials” has been excluded and changed.

  1. The authors must eliminate bio-composites, because only a fraction of the compound is bio.

Answer: We are thankful for drawing our attention to this problem. Therefore, there is no statement that we developed “bio-composite” or “bio-material” any more.

  1. The authors must eliminate healthcare, although the material can be applied in this sector, the article does not evaluate its properties in these applications.

Answer: We agree with this comment. Therefore, the “healthcare” potential of the described material has been eliminated. Nevertheless, we decided to mention about this possible application while underlining that the further investigation is necessary: Moreover, the combination of unusual cellulose features with the characteristics of the ethylene-norbornene copolymer may lead to the polymer composite healthcare applications development. The mentioned polymer matrix is widely known for its resistance to aqueous and polar organic media, as well as good biocompatibility, easy processing and forming. Moreover, the study performed by Wang et al. [59] indicate that MA containing materials have no significant toxic effect regarding the growth of both S. aureus and E. coli. Therefore, plant fiber-based polymer materials described in this research might be possibly applied in medical and healthcare applications, e.g., drug delivery, wearables, different medical devices, pharmaceutical blisters and trays. Nonetheless, further research regarding the toxicity of maleic anhydride should be carried out in the future.

  1. Line 42. The authors say "In the following research, an application of fully biodegradable natural fibers is presented". First, using the term fully biodegradable natural fibers is redundant, by definition, natural fibers are fully biodegradable. Secondly, this sentence does not provide any information since the authors would rather be referring to paper production.

Answer: We agree with this statement. Therefore, the sentence has been deleted.

  1. Next, the authors make a state of the art on methods of compatibilization natural fibers - plastic matrix. However, they omit the most used method and with better results, the use of coupling agents (plastic polymer with maleic anhydride). The authors should extend the introduction in this sense.

Answer: We are sorry for omitting this method in the introduction. It is true that this is incredibly relevant. Therefore, Introduction has been improved: This article concentrates on the process of cellulose maleinization and its development. It is opposite to the commonly used process of polymer matrix MA grafting and, then, mixing with cellulose. In this study, an alternative approach, which aim is to employ MA not as a direct coupling agent between cellulose and polymer matrix, but as a factor altering the cellulose surface properties, (e.g., hydrophobicity, surface area) is presented. Nonetheless, it should be highlighted that employment of MA modified PE or PP (systems similar to the ethylene-norbornene copolymer) filled with natural fibers provides polymer composites of a good mechanical characterization [47–50].

  1. The authors justify the novelty and relevance of the article on the basis that this method allows the use of natural fibers in the production of composite materials. However, there are numerous works where coupling agents type MAPP or MAPE are used, which are much easier and have better performance in terms of improvement of mechanical properties.

Answer: We are thankful for this comment. Regarding the Reviewer’s opinion we have developed the justification: Therefore, this research is a broadening of the previously presented topic and successfully presents the outcome of the improved cellulose treatment process. Here, a combined cellulose modification approach have been introduced – physical treatment was combined with a chemical one. Hybrid chemical modification consists of two steps: solvent exchange (with ethanol either hexane) and chemical grafting (maleic anhydride - MA). MA has been used in order to change the surface characterization of the cellulose fibers and to improve the filler-polymer matrix adhesion. Its positive effect on the composite mechanical properties was described in literature [53–55]. Thanks to the performed complex modification process, cellulose is firstly pre-treated in order to change its physical features, such as molecular packing, pore sizes and hydrogen bonds organization, which may lead to the reactivity altering [56]. Moreover, considering the effect of coupling agents on the properties of polymer composite properties we have presented the results of different researches in Table 3.

  1. Although the authors have previously published the method of modifying the surface of the fibers, they must explain step by step the modification of these including all the conditions of the process.

Answer: We are thankful for this comment. We believe that now it easier for reader to understand the process of cellulose modification and its steps: The major advantage of the presented modification approach is the fact that commonly known chemical treatment of cellulose fibers with MA was broadened with the physical modification (pre-swelling) in media of various polarity and structure (ethanol and hexane) enabling more efficient treatment and improvement in polymer composite performance. The presented study covers a few problems: effect of the solvent exchange (different polarity) on the properties of cellulose fibers, impact of the dehydration process on the filler modification process, the combined influence of drying, solvent exchange and MA grafting on the properties of both the cellulose and filled polymer composites. Solvent exchange has been carried out as follows: dispersion mixing (cellulose [g] to solvent [mL] ratio – 1:10; magnetic stirrer, 8 h, room temperature) and its conditioning for 16 h in ambient conditions (no stirring). Then, the solvent was distilled in a vacuum rotary evaporator at 40°C (60 rpm, 100 mbar for ethanol and 250 mbar for hexane). Modification with maleic anhydride: MA solution with dispersed cellulose (cellulose [g] to acetone [mL] ratio – 1:10, cellulose [g] to MA [g] ratio – 4:1, 2 h, 40°C, stirring).  Acetone was removed with the vacuum distillation process (40°C, initial pressure 200 mbar). Further drying: vacuum oven (100°C, 440 mbar, 4 h). Between the steps of solvent exchange and modification with MA, cellulose fibers have been stored at 70°C and then after finishing whole modification process – at 40°C (before incorporation into the polymer matrix). Summary of the carried out experiments are presented in Table 2.

  1. The authors do not specify the meaning of H and E in table 2.

Answer: We are terribly sorry for this mistake. It has been corrected.

  1. Line 135. What does phr mean? Are they percentages by weight? In volume? What do 8, 16 and 40 answer? Why have these percentages been chosen?

Answer: phr states for ‘per hundred rubber’ and it is commonly used in describing the polymeric mixture content. Nonetheless, it has been changed to wt%.

  1. Lines 137-138. The authors say "". In the mill they orient fibers once the composite is produced?

Answer: We believe that this misunderstanding is a confusion caused due to not clearly enough described process. Therefore, it has been improved: Modified cellulose fibers were dried for 24 h at 100°C before incorporation into the polymer matrix. The polymer blends of cellulose (7, 14, 35 wt%) and ethylene-norbornene copolymer were prepared in a laboratory micromixer (Brabender Lab-Station from Plasti-Corder with Julabo cooling system, Duisburg, Germany) at 110°C (30 min, 50 rpm). For orientation of the fibers, the mixture has been put between two-rolling mills (100x200 mm rolls, 20–25°C, friction of 1:1.1, 45 seconds). Then, the it was compressed in a hydraulic press at 160°C (10 min, 125 bar) between two steel molds and Teflon sheets – the plates were obtained. We do not orientate the fibers once the composite is produced – it is an attempt to orientate the fibres at the stage of processing.

  1. The authors must at least explain on what are based the methods used (gravimetry, etc...) and under which norms the tests have been conducted.

Answer: The norms has been introduced where necessary. Nevertheless, we do not fully understand what is the additional problem with the description as we did not use gravimetry in this research – only tensile tests and dynamic mechanical analysis. These are commonly used techniques regarding polymer composite properties investigation. Nevertheless, we have revised the description once more.

  1. The authors should know the importance of the type of fibers used. A chemical and morphological characterization of Arbocel fibers is essential to understand their behavior in the modification and inside the composite material. They should include: Lignin content, hemicelluloses, cellulose, average length, average diameter, length distribution, polarity, etc.

Answer: We are thankful for drawing our attention to this problem. The available details regarding the cellulose employed in this research were given in the Materials section: Arbocel UFC100 Ultrafine Cellulose for Paper and Board Coating from J. Rettenmaier & Soehne (Rosenberg, Germany) was employed in the following research (fiber length: 8-14 µm, density: 1.3 g/cm3, pH: 5.0–7.5).

  1. The authors base the article on the modification of the fibers and do not present any results: polarity variation, cationic demand, IR, etc.

Answer: As it was mentioned, this research is a part of the bigger project, as mentioned in Section 2.2.: Some properties of polymer composites filled with hybrid chemically modified cellulose fibers, as well as the treatment process effect on biopolymer characteristics, are described in the separate scientific articles:

  • moisture content effect on properties of cellulose fibers [56] and polymer composites [60],
  • hybrid chemical modification process – biopolymer hydrophobisation [61],
  • thermal behavior of hybrid chemically modified cellulose filled polymer blends [62].

Addition information could be found in different articles debating about the cellulose fibres modified via this approach.

  1. The graphics of the figures are extremely difficult to read.

Answer: We have tired to make the images sharper and change the figures in order to improve their readability. Nonetheless, we and different Reviewers find it clear and well presented.

  1. The incorporation of cellulose fibers reduces the properties of the matrix, which indicates an ineffective modification of the fibers.

Answer: We are grateful for this comment as we think that this was not sufficiently described before: Data presented in Fig. 1a-d indicate that carried out cellulose treatments exhibit a varied effect on the mechanical properties of prepared composite samples. It could be observed that many treatments have led to the decrease in composite specimen mechanical performance. According to literature, it could be caused by insufficient cellulose hydrophobization with the employment of MA as it most commonly used in the role of coupling and not hydrophobizing agent [54,68,69]. Nonetheless, in some particular cases it was possible to regenerate the performance of pure polymer matrix or even to create a material of an improved tensile strength or elongation at break. This is very promising and not common according to the information given in literature [52,70–72] and highlights the significance of cellulose structure altering during the solvent exchange step [56]. Moreover, we would like to underline that this article covers the whole spectrum of different modification combinations and not each of them has given a satisfying result. Nevertheless, we decided to show the whole spectrum. Furthermore, some samples are exhibiting really good properties which has been underlined by dedicating them the following section:  3.3. Comparison of the obtained results with literature. Here, the behaviour of the sample filled with ND/MA/1/E specimen is deeply described regarding different content of cellulose fibres loading and the reinforcing effect of the cellulose is revealed.

Reviewer 3 Report

The need for eco-friendly materials is explained as well as the cellulose polarisation as an obstacle for its incorporation of polymer matrices. Available surface modification methods of cellulose are described, highlighting the grafting as the most promising one.

Need for current research is justified by the need of improved properties of modified-cellulose loaded ethylene-norbornene copolymer composites, which is also the work of the same authors. The only innovation proposed in the introduction is the solvent exchange before chemical process. Without looking on the “big picture” of all related  papers published by the same authors recently, it seems low novelty potential for this individual manuscript.

  1. More justification of using specifically maleic acid for grafting the cellulose surface is needed in the introduction. What are the advantages of using cellulose maleinization? Which other cellulose like polymers have been grafted and investigated/published using this method?
  2. Is there any proof about non-toxicity of cellulose, when it is modified via maleinization? It should be assessed before proposing for application in medical or healthcare applications.
  3. Graphical abstract of manuscript would improve the manuscript.
  4. It is great to have summary table of performed modifications, however there is no detailed description of cellulose modification method. Reaction conditions, amounts of used chemicals etc should be added. In my opinion it is not enough to give reference to authors’ previous works. Method must be briefly described also in the current manuscript.
  5. The question “What was the size of used cellulose fibres?” occurred during reading the first part of manuscript and it was answered later in Table 3. Considering small sizes of cellulose particles, why do you still call them fibres instead of cellulose microparticles or microcrystalline cellulose (MCC)? Measurements of crystallinity would reveal, if it is MCC. Were there some visualisation methods used (SEM, AFM) in order to reveal shape of cellulose particles?
  6. Line 134 Methodology of cellulose fibers drying process has been given, however there is no details about incorporation of non-dried fibres in polymer matrix. Was it the same procedure described in Lines 135 – 141. Did non-dried fibers act the same way as dried in the composite forming process?
  7. It is confusing – is the drying step described in Line 134 the same included in Table 1? So when exactly the celullose was dried? Before/after the modification AND before incorporation into polymer matrix? Does it mean that some of samples were dried several times during the whole process? Detailed description of modification process, even graphical image with scheme, would make the methodology more clear and there would be no place for confusion of the reader.

It should be taken into account that if cellulose material was delivered as dry powder, it has already had some wetting- drying cycles. In order to completely exclude hornification effect, never-dried fibres should be used.

  1. Figure 1. The images give information about neat cellulose as better filler than modified one in some cases? Could you please explain this fact?

In overall, several theories have been proposed and applied to purposefully explain debatable results of effect of modified cellulose on the properties of polymer matrix. Seems quite challenging, however proposed filler efficiency factor and Payne effect explain the results more understandable.

Significant work done on explanation and interpretation of numbers obtained, which resulted in choosing the most promising modification method for further research.

The justification for the need of the particular research  should be improved in the text (introduction), considering this paperwork as individual and not only emphasize it as the continuation of previous work done. The fact that the reader may do not have previous knowledge or information about the specific topic must be considered as well, giving more overall information in the text, not only by referring to previous papers.

Author Response

Institute of Polymer and Dye Technology

Technical University of Lodz

90-924 Lodz, ul Stefanowskiego 12/16, Poland

Tel.: +48 42 631 32 23, Fax: +48 42 636 25 43

November 12, 2020

Materials

Dear Professor,

We are resubmitting our revised paper entitled Mechanical Characterization of Sustainable High-Performance Polymer Bio-Composites for Potential Healthcare Applications (then: Cellulose Modification for Improved Compatibility with the Polymer Matrix: Mechanical Characterization of the Composite Material) by Stefan Cichosz, Anna Masek with a request to reconsider it for publication in Materials.

We have carefully considered the Editor and Reviewers' comments. The manuscript was revised exactly according to these comments. The list of responses to the reviewer’s comments and corrections made in the manuscript is attached.

The manuscript has not been previously published, is not currently submitted for review to any other journal, and will not be submitted elsewhere before a decision is made by this journal.

For correspondence please use the following information:

corresponding author: Anna Masek

Institute of Polymer and Dye Technology

Technical University of Lodz

90-924 Lodz, ul Stefanowskiego 12/16, Poland

Tel.: +48 42 631 32 93

Fax: +48 42 636 25 43

e-mail: anna.masek@p.lodz.pl

Yours sincerely,

Ph. D., D.Sc. Anna Masek

All changes are marked with a green colour through whole manuscript.

Reviewer #3

The paper is focused on the mechanical properties of plant fiber-filled polymer composites. The topic falls within the scope of the journal. The MS can be published after the following revisions.

The comments are listed below:

  1. Conclusions paragraph should be shortened.

Answer: We are thankful for this comment. We have shortened this section.

  1. Some examples of stress vs strain curves should be presented.

Answer: We have presented the stress vs strain curves in the Supplementary File: Mechanical properties as tensile strength (TS), elongation at break (Eb) and Young modulus values at elongation equal 100%, 200%, 300% (SE100, SE200, SE300) have been determined with the use of Zwick-Roel Z005 (Ulm, Germany) measuring device. Tests were carried out on dumbbell shape, approximately 1.5 mm thick and 4 mm width specimens. According to: PN-ISO 37:1998 standard. The exemplary stress-strain curves available in Supplementary Materials (Fig. S1).

  1. I suggest to estimate the stored energy up to breaking by the analysis of stress vs stain curves.

Answer: We are grateful for this idea and we find it as a a valuable tip for the future. Nevertheless, this time we are not able to calculate the stored energy as we possess only graphical files of stress-strain curves and test reports with values presented in the manuscript. As we did not think about estimating stored energy, the numerical files were not exported and, unfortunately, it is impossible right now. Nevertheless, it is a really valuable comment.

  1. Introduction might be improved by quoting recent articles [New Journal of Chemistry 2019, 43, 10887-10893] and reviews [Carbohydrate Polymers 245 (2020) 116502] focused on the development of bio-composite materials for healthcare purposes.

Answer: We are grateful for providing these articles enriching our references.

Moreover, we would like to underline that the whole manuscript has been revised carefully considering some language mistakes and appropriate meaning. Changes have been marked throughout the

Round 2

Reviewer 1 Report

The paper is suitable for publication in the present form.

Reviewer 2 Report

The reviewer would like to thank the authors for considering their comments. In my opinion the modified changes have significantly increased the quality of the article.

This manuscript is a resubmission of an earlier submission. The following is a list of the peer review reports and author responses from that submission.

Round 1

Reviewer 1 Report

Review report

Manuscript ID: polymers-946728

Title:  Mechanical Characterization of Sustainable High-Performance Polymer Bio-Composites for Potential Healthcare Applications

The article presents the two-step modification process of cellulose fibres and discusses the effect of these modified fibres onto the mechanical properties of ethylene-norbornene copolymers. The modification process of cellulose fibres and the composite preparation process described in this article have already been published by the same group (ref 44, 45 and Polymer Bulletin (2019), volume 76, pages 2147–2162). Surprisingly, the authors did not mention this in the “Introduction”. However, there are some points which are not clear.

The comments are:

  1. Cellulose modification with MA is widely studied. The authors should highlight the difference between reported works on MA-modified cellulose and the present work. The authors should clarify the rationale behind this complex modification process. Also mention the drawbacks of previously reported solvent exchange and chemical modification methods of cellulose by MA.

    Also, the authors should discuss the reported works on cellulose reinforced ethylene-norbornene copolymers

  2. Section 2.1.2, Page 3, line 122-124: “The regular surface modification of cellulose fibers was broadened with the solvent exchange– from water to ethanol either hexane”.

Section 2.1.2, Page 3, line 127-128: “Moreover, cellulose fibers have been dried (24 h, 100°C; crystallizer 70x40 mm) either not dried before the hybrid chemical modification process. – Please rephrase the sentences, the expression is not clear.

  1. It is not clear why did you dry the fibres before chemical modification since drying of cellulose fibres lead to hornification and this might make the modification process difficult.

  1. What is the reason of using the solvent exchange step before chemical modification with MA? Did you use any acid catalyst during chemical modification? How can you confirm the esterification reaction is happening there or there is any physical interaction between MA and cellulose?

  1. It is not clear from the manuscript if there is any chemical interaction between filler and matrix. Did you perform FTIR of the composites?

Overall, the work is lack of novelty. The hypothesis of the work and expression are not very clear throughout the manuscript. The article is not ready to recommend for publication at this stage.

Reviewer 2 Report

In this article, authors have reported on the development of sustainable composites.

Following minor corrections are needed:

-Abstract should be re-written. Especially the first sentence needs good wording.

-Revise the article for language correction. For example "Raising environmental " should be rising

-Figure 1 can be removed

-DMA properties should be explained in more detail

-Intrdouction should be strengthened by citing relevant articles such as Polymers 202012(7), 1472; Chemical Reviews 120 (17), 9304–9362